# TabDEG: Classifying differentially expressed genes from RNA-seq data based on feature extraction and deep learning framework

**Sifan Feng, Zhenyou Wang, Yinghua Jin***, Shengbin Xu

School of Mathematics and Statistics, Guangdong University of Technology, Guangzhou, Guangdong, China

* jyh@mail.ustc.edu.cn

**Data Availability Statement:** The code.zip file is the code involved in the article, and the data.zip file is the data used in the experiments in the article,

## Abstract

Traditional differential expression genes (DEGs) identification models have limitations in small sample size datasets because they require meeting distribution assumptions, otherwise resulting high false positive/negative rates due to sample variation. In contrast, tabular data model based on deep learning (DL) frameworks do not need to consider the data distribution types and sample variation. However, applying DL to RNA-Seq data is still a challenge due to the lack of proper labeling and the small sample size compared to the number of genes. Data augmentation (DA) extracts data features using different methods and procedures, which can significantly increase complementary pseudo-values from limited data without significant additional cost. Based on this, we combine DA and DL framework-based tabular data model, propose a model TabDEG, to predict DEGs and their up-regulation/down-regulation directions from gene expression data obtained from the Cancer Genome Atlas database. Compared to five counterpart methods, TabDEG has high sensitivity and low misclassification rates. Experiment shows that TabDEG is robust and effective in enhancing data features to facilitate classification of high-dimensional small sample size datasets and validates that TabDEG-predicted DEGs are mapped to important gene ontology terms and pathways associated with cancer.

## Introduction

NGS(Next-generation Sequencing) is a high-throughput sequencing method used to analyze the transcriptome and genome of a species. It is considered revolutionary and widely used in genetics [1]. Many gene sequencing technologies, such as whole-genome DNA sequencing and total RNA-sequencing, utilize NGS. Total RNA sequencing can detect multiple forms of non-coding RNA and then overcome limitations of traditional microarray experiments [2–4]. RNA sequencing (RNA-Seq) technology, with its own merits mentioned above, is widely used in gene expression analysis. It can be used to explore the evolution of gene expression [5] and assess the impact of gene expression levels in medicine [6]. Additionally, RNA-Seq enables certain applications that are not possible with microarray experiments. For instance, gene differential expression profile based on RNA-Seq in disease tissues (such as cancer cells) is used to

and these have now been uploaded to: https://github.com/xueyupi/my_tabdeg.git.

**Funding:** This work was supported by a Natural Science Foundation of Guangdong Province grant (2023A1515012891 awarded to ZW). The funders had no role in study design, data collection and analysis, decision to publish, or preparation of the manuscript.

**Competing interests:** The authors have declared that no competing interests exist.

obtain a complete transcriptomic profile for discovery of new genes and information about their function, mechanism of action, and pathways [7]. With the market availability of NGS platforms, RNA-Seq has become an attractive alternative method to traditional ones for detecting differentially expressed genes (DEGs). By conducting differentially expressed (DE) analysis, researchers can identify genes that undergo changes in biological patterns between health and disease conditions. This can help prioritize these condition-specific genes as potential biomarkers for a particular disease [8].

RNA-Seq data is mapped to a reference genome and summarized as "reads". Existing methods for classifying microarray data [9–11] cannot be applied to RNA-Seq data due to potential discrete distributions, such as Poisson and Negative Binomial [12, 13]. Thus, several statistical methods have been proposed for RNA-Seq data analysis [14]. Normalization is the first step in RNA-Seq data analysis to eliminate variations caused by different numbers of reads from different samples. "Sequencing Depth (SD)" [15], proportional to the total number of reads, is estimated and used to divide individual counts for normalization. Accurate normalization ensures unbiased interpretation of RNA-Seq data.

There are two problems with current pre-normalization strategies in RNA-Seq data analysis: different methods may estimate SD differently, and there are no systematic guidelines for selecting the best method [16]. Additionally, the assumption that most genes are not differentially expressed (DE) between samples in RNA-Seq analysis may not always hold true, such as in cancer samples where most genes may be DE [17]. To address these issues, scale-invariant (SI) [18] methods have been proposed, which give consistent results despite differences in SD estimates. However, SI approaches may not be applicable to all tasks, and some non-SI methods can result in biased results [19, 20]. This can lead to high rates of false positives and false negatives when predicting DE genes in RNA-Seq data analysis.

## The challenges and solutions of deep learning-based DEGs identification

In recent years, many machine learning (ML) methods have been developed for gene expression classification. For example, support vector machines (SVM) use mutual information and Hine to classify genes to distinguish between colon cancer patients and healthy patients [21]; logistic regression (LR) has also been used to classify different types of cancers and normal tissues [22]; A randomized rest-based approach to classify genes in microarray data was also proposed [23]. However, these ML methods require prior knowledge about gene features for classifier training.

In the particular case of high-throughput genomics, with the increase of gene expression data, deep learning (DL) has been shown to capture internal structural features of biological data and extract high-level abstract features from high-dimensional sequencing or expression data, thereby improving the performance and interpretability of traditional ML methods [18]. DL includes a series of techniques such as multi-layer neural networks (MLNN), convolutional neural networks (CNN), deep autoencoders (AE), and recurrent neural networks (RNN), which have achieved excellent performance in fields such as computer vision, pattern recognition, and natural language processing [24]. However, applying DL to RNA-Seq is still challenging because the sample size is small and lacks appropriate labels compared to the huge number of genes [25, 26]. To address the imbalance between sample size and features (genes), two types of solutions are proposed in the literature: data augmentation (DA) and transfer learning (TL) strategies. It should be noted that compared to TL, DA is more suitable for small-sample datasets, as it can enhance model robustness and generalization ability.

Data augmentation (DA) is a process of enlarging the training dataset by generating pseudo data equivalent to more data without significantly increasing the amount of actual data, using

various methods and procedures. Generally, DA can be classified into supervised and unsupervised methods. Supervised DA can be further divided into single sample DA and multiple sample DA, while unsupervised DA is divided into generating new data and learning augmentation strategy. For instance, affine or perspective transformations can be applied to images [27–29], as well as more advanced techniques based on variational autoencoders [30] and generative adversarial neural networks [31]. In experimental settings, DA has biological significance in that it allows for the generation of new data points with varied biologically relevant features, enabling better study of biological variability under different conditions. These new data points can be utilized to enhance machine learning models' accuracy, classification and prediction results, as well as validate and support biological hypotheses [32]. Moreover, DA can aid in addressing noise and uncertainty in experiments, thereby improving the quality and reliability of data. By leveraging data augmentation, researchers can better utilize the original data and obtain more meaningful and accurate information from it, thereby enhancing our understanding and interpretation of biological phenomena and problems [33].

## Selecting the suitable deep network model

In addition, there are limitations to commonly used techniques for neural networks in gene expression data due to its unstructured nature. In a recent study, multilayer perceptron (MLP) and CNN were applied in combination with linear discriminant analysis (LDA), logistic regression (LR), plain Bayesian (NB), random forest (RF), and support vector machine (SVM) methods to predict disease staging or distinguish diseased samples from normal samples by comparing tests on more than 30 gene expression genes [34]. The results showed that CNN were one of the worst-performing methods in most of the analyzed datasets. The reason for this outcome may be that the effectiveness of CNN depends on how the convolution filter exploits the local motif patterns present in the analyzed data. In the image domain, neighboring pixels do not share information independently, so local patterns can be extracted through the convolution layer. However, gene expression data lacks local patterns because neighboring genes in the same sample are considered to be independent. Hence, gene expression samples do not exhibit "spatial coherence", and it is the "local" regions of genes in the sample that do not share similar information. Therefore, if pixels in an image are rearranged, their relative positions change, which may negatively affect the feature extraction and performance of CNNs [35]. RNN, as a class of neural networks for sequential data (e.g. time series and text series), is very effective for data with sequential properties [36], but less effective for gene expression types. In order to take advantage of DL's ability to process large datasets and reduce the need for feature engineering by other machine learning methods, a Google experimental team proposed a new high-performance and interpretable canonical deep tabular data learning architecture, TabNet [37]. TabNet could enhance the ability of the end-to-end learning process to efficiently encode tabular type data such as gene expression data. End-to-end learning is a learning paradigm provided by deep learning, where the entire learning process is left entirely to the DL model to learn the mapping from the original data to the desired output directly [38]. This approach circumvents the inherent pitfalls of multiple modules and could reduce the complexity of the operational process.

It is possible to create a model based on TabNet architecture that uses trained features to identify DEGs in large datasets and predict upregulating (UR)/downregulating (DR) directions. To our knowledge, there are no end-to-end models in existing literature based on attentional mechanisms for deep neural network (DNN) models combined with DA methods to predict DEGs and determine their direction of action by training gene expression data. Considering the excellent performance of TabNet network on tabular data, we propose a robust

model (TabDEG) in this paper, which combines DA methods to classify DEGs and predict UR/DR directions from RNA-seq cancer datasets with higher accuracy than other ML approaches.

## Materials and methods

### Data collection and preprocessing

Cancer datasets used in this paper are downloaded from UCSX Xena Data Browser(the UCSX Xena Data Browser serves as a data browser that provides access to and analyzes TCGA data, https://xenabrowser.net/datapages/) and they belong to the type of pan-cancer. This type of cancer dataset includes *11K RNA-Seq gene expression samples from 10 different tumor types and is converted to RSEM values by transformation $log2(TPM + 0.001)$. The TCGA dataset selected in this paper comes from a variety of samples from patients with different regions, races and clinical characteristics, and thus contains different genomic information, which helps to reduce group bias and improve the representativeness of the results. Meanwhile, different tumor types may have different characteristics, development patterns, and treatments, so by using datasets of multiple tumor types, the over-reliance on specific tumor types can be reduced, and the generalization ability of the algorithm can be improved, making it more robust in dealing with unknown or novel tumor types. The ten datasets presented in Table 1 have different sample size and each sample contains expression values for 60,488 input variables (transcripts).

The data pre-processing steps are listed as follows (using COAD dataset as an example):

1. The dataset from TCGA is prepared and represented as an expression matrix with genes as rows and samples as columns. For all 60,488 RNA transcripts, all samples are screened from the tumor and normal groups, i.e. all columns with "01A" and "11A" in their names.

**Table 1. Values of hyperparameters used in TabDEG model.**

| Hyperparameters | Train | Test |
| --- | --- | --- |
| max_epochs | 200 | 200 |
| patience | 50 | 50 |
| batch_size | 512 | 512 |
| virtual_batch_size | 64 | 64 |
| drop_last | False | False |
| loss_fn | loss_fn | loss_fn |
| learning_rate | 5e-4 | 5e-4 |
| weight_decay | 1e-5 | 1e-5 |
| n_d | 8 | 8 |
| n_a | 8 | 8 |
| n_step | 1 | 1 |
| lambda_sparse | 0 | 0 |
| optimizer_fn | torch.optim.Adam | torch.optim.Adam |
| optimizer_params | dict(lr = 2e-2,weight decay = le-5) | dict(lr = 2e-2,weight decay = le-5) |
| mask_type | "entmax" | "entmax" |
| scheduler_params | dict(milestones=[50, 100, 150],gamma = 0.95) | dict(milestones=[25, 50, 100, 150],gamma = 0.95) |
| scheduler_fn | torch.optim.lr_schedulerMultiStepLR | torch.optim.lr_schedulerMultiStepLR |
| verbose | 10 | 10 |
| eval_metric | LogLossMetric,SmoothedLogLossMetric | LogLossMetric,SmoothedLogLossMetric |

2. Import the gene annotation file gene_length_Table, which contains a total of 56,716 transcript annotation information. And the annotation information of only 19,627 coding RNAs is needed in this paper. The ENSEMBL identifier of RNA transcript is used to intersect TCGA dataset and each row in TCGA dataset is labelled with the corresponding gene name.

3. Import the "tidyverse" package and then start pre-processing the gene expression data. RNA transcripts with an expression level greater than 1 are screened out and a total of 17983 RNA transcripts are screened. The screen-obtained datasets are divided into $T$ and $N$ according to its column names (see Step 1 for details), which are used as control conditions in the next step.

4. After importing "DESeq2" package, the function DESeqDataSetFromMatrix() is used to construct a DESeqDataSet(dds) object from the expression matrix and sample information. After constructing the dds object, DESeq() is used to perform variance analysis.

5. Since in RNA-seq data, different sample conditions have different sequencing depths and RNA compositions, which may negatively affect downstream analysis, the function estimateSizeFactors() is used to calculate the normalization coefficient 'sizeFactor' in the process of differential analysis. Then estimateDispersions() is used to estimate the degree of gene dispersion. Finally, statistical tests are performed using nbinomWaldTest() and the results of variance analysis could be obtained by calling results().

6. In the result data of difference analysis, the screening conditions are set as abs(log2FoldChange) > 1 and padj < 0.05, and genes satisfying these two conditions are treated as DEGs. Non-DEGs are labelled as "2".

7. Finally, DEGs are marked as UR/DR directions based on logFC thresholds. If logFC > 1, DEGs are treated as UR genes and marked as "1"; and if logFC < -1, DEGs are treated as DR genes and marked as "0". LogFC > 1 indicates at least a 2-fold change in the expression level of a gene between two conditions, and such a change is considered to be relatively large to provide a higher significance difference and thus more likely to reflect biological importance. In many related studies [39], logFC > 1 is considered a reasonable threshold for screening out DEGs of functional and biological importance.

## Data augmentation

In the introduction, it is pointed out that applying DL to RNA-Seq data is still very challenging due to the small sample size (n) relative to the number of genes (g). To address this problem, we adopt a DA method to simulate biological changes and generate data similar to natural observations [33], partially alleviating the huge imbalance between sample size and feature (gene) number in RNA-Seq data and improving the recognition and classification ability of DL models for complex and variable biological data. DA can be achieved by combining different dimensionality reduction and clustering methods. In this paper, we use PCA, UMAP, and K-means methods. PCA is a common dimensionality reduction method that finds the main components of data by seeking directions that maximize variance, but it may ignore variance in other directions and thus overlook fine structures [40]. In contrast, UMAP aims to preserve the topological relationships of adjacent samples in the data, namely finer local structures [41]. PCA and UMAP have different characteristics and can preserve different types of structures in the data. Combining them can more effectively extract features from the data. K-means

clustering is a popular unsupervised learning method that has high accuracy in small sample clustering and can optimize iterative functions to iteratively correct and prune clusters obtained to determine the clustering of certain samples [42]. By using these three methods simultaneously, we can efficiently perform data augmentation while minimizing accuracy loss.

In experiments, we use statistical knowledge and ML techniques to expand effective features of TCGA data sets that have been preprocessed to improve the performance of DL models. Specifically, we classify the features of the preprocessed TCGA data set by keyword into two groups, *T* and *N*, and then apply factor analysis (FA), UMAP, and K-means methods to each group. Finally, we extract the information of the two groups of features and merge the results to obtain the original data set. The detailed workflow of the entire process can be seen in Fig 1.

**Factor analysis.**   FA is a correlation-based data analysis technique and is usually used to do dimensionality reduction in multidimensional data analysis. In essence, FA could be used to explore some kind of structure hidden behind multidimensional data with a large number of observations, with aim to find common factors for variation in a set of variables and to group variables with same nature into a factor [40]. There are many types of FA and the method of Principal Component Analysis (PCA) is used in this paper.

In a TCGA dataset, assume that genes are denoted as $(X_1, X_2, \ldots X_M)$ and $M$ represents the number of genes. For each gene $X_i(i = 1, 2, \ldots, M)$, the corresponding sample is denoted as $(x_{i1}, x_{i2}, \ldots, x_{iN})$, that is, the feature corresponding to each gene has $N$ columns. Thus, RNA--Seq data containing $N$ samples of $M$ genes could be recorded as an expression matrix $X$ with dimension $M \times N$. The empirical mean of each dimension (genes) is denoted as $u_i(i = 1, \ldots, M)$ where $u_i = \frac{1}{N}\sum_{j=1}^{N} x_{ij}$. And the deviation matrix $B = X^T - hu^T$ from the mean is calculated by the empirical mean vector $u = (u_1, u_2, \ldots u_M)^T$ where $h$ is a column vector of dimension $N \times 1$ with all entries being 1. Further calculate the covariance matrix $C = \frac{1}{N-1}B^*B$ of the above deviation matrix where $*$ is the conjugate transpose operator. Finally, the covariance matrix $C$

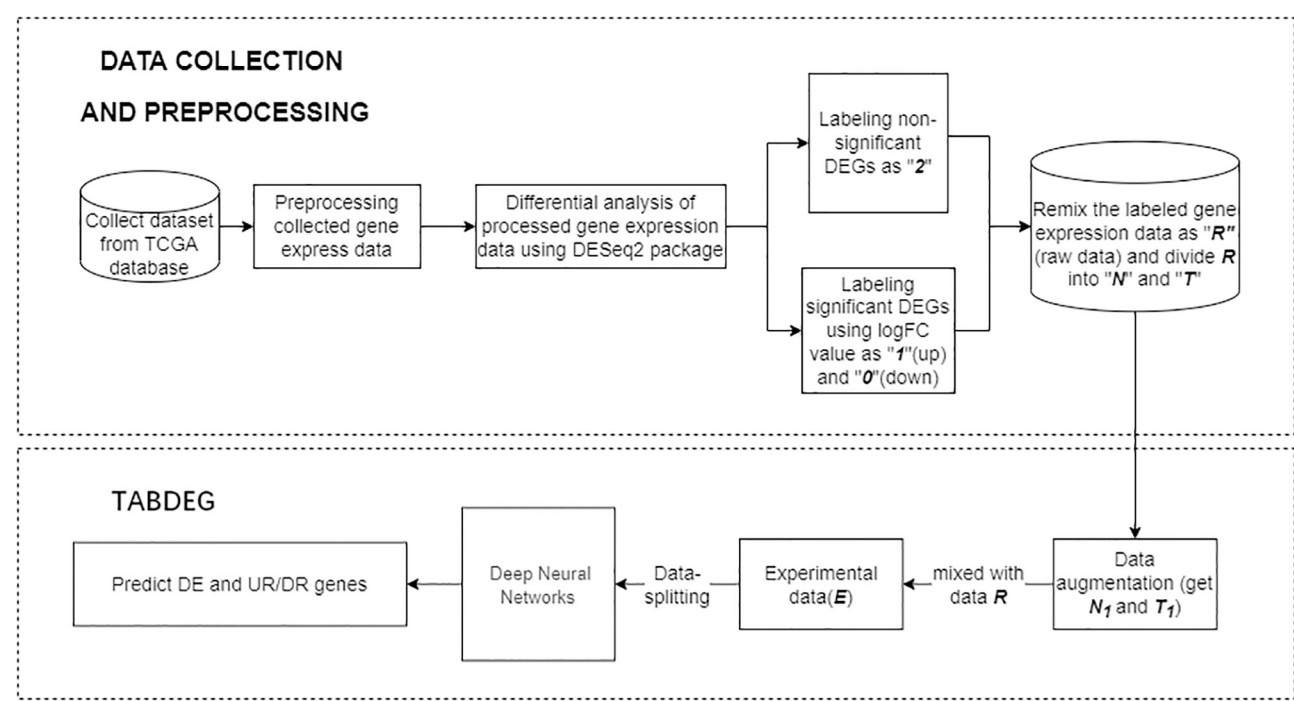

**Fig 1. Workflow of TabDEG.**

is decomposed as:

$$C = VDV^T$$

where $V$ is a $M \times N$ dimensional matrix composed of $N$ eigenvectors for $C$ and $D$ is a diagonal matrix with dimension $N$ composed of $N$ eigenvalues for $C$. All eigenvectors and eigenvalues in $V$ and $D$ are paired in decreasing order for eigenvalues, that is, the $i - th$ eigenvalue corresponds to the $i - th$ eigenvector. The appropriate top $p$ feature vectors are chosen and treated as the basis for new dimensions and these new dimensions enable meaningful data augmentation.

**UMAP.**  UMAP is an algorithm for dimensionality reduction by mapping high-dimensional probability distributions to a low-dimensional space [41]. The algorithm is mainly based on the theory of refinement and topological algorithms, which can preserve more global structure and continuity of data subsets, and has superior runtime performance and better scalability. In addition, UMAP has no computational restrictions on the embedding dimension, so it can be used as a generalized machine learning dimensionality reduction technique [43]. In this paper, UMAP method is used to process the pre-processed TCGA dataset with aim to obtain "new" features and this process mainly consists of two steps:

(i)Learning about flow structures in high-dimensional space [41]. In the exploratory analysis of the TCGA dataset, UMAP represents data using a weighted graph called "fuzzy surface complex", and finds nearest neighbors by expanding the radius of each point, ensuring a balance between local and global structure.

(ii)Finding a low-dimensional representation of the manifold considered [41]. The learned manifold is projected onto a low-dimensional space, and the standard Euclidean distance in the globally coordinate system is used to set distances on the manifold. By selecting and controlling the minimum point distribution to avoid overlap, the cross-entropy cost function is minimized to find the optimal weight of edges in the low-dimensional representation, and stochastic gradient descent is used to perform the minimization process.

$$CE = \sum_{e \in E} \{ w_h(e) log \frac{w_h(e)}{w_l(e)} + (1 - w_h(e)) log \frac{1 - w_h(e)}{1 - w_l(e)} \}$$

For edges() in the set $E$(i.e. all edges found in step 1), the first term in above formula would act as an "attractor force" as long as there is a large weight associated with the high-dimensional case. The reason may be that this term will be minimized in the largest possible case which would occur when the distance between points involved is as large as possible. When a high-dimensional weight is small, the second term acts as a "repulsive force". The reason may be that this term will be minimized by making as small as possible. Finally, an array containing the coordinates of each data point is obtained in the specified low-dimensional space. The complete UMAP workflow is showed in Fig 2.

**K-MEANS.**  In the process of clustering analysis, the number of clusters $K$ is generally determined in advance so that the samples in the same clusters are distributed as closely as possible and the distance between clusters is as large as possible. Almost all algorithms for clustering analysis are designed to divide the data analyzed into independent sets of data samples such that the variances between sets of clusters are equal and this process is mathematically described as minimizing the sum of squares within clusters [42]. As an unsupervised clustering algorithm, K-means is relatively simple to implement and has good clustering effect so that it is widely used.

There has been perfect strategy to chose the number of clusters $K$. The Calinski-Harabasz index is essentially the ratio of the inter-cluster distance to the intra-cluster distance and the overall calculation process is similar to the variance calculation, so it is also referred to as the

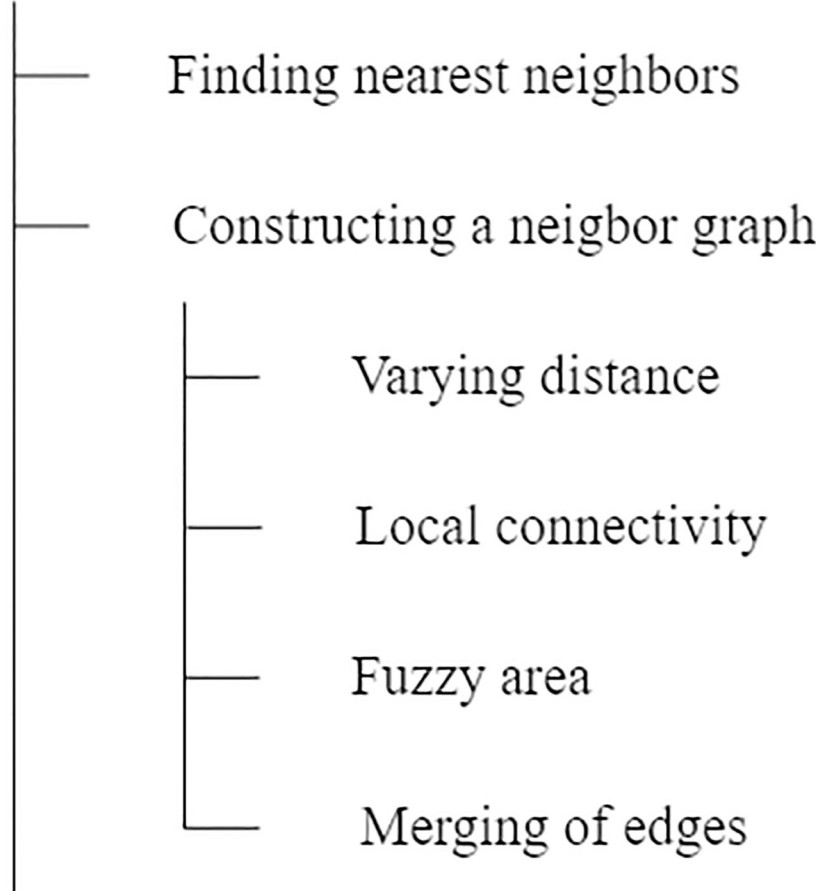

# Step 1 - Learning the manifold structure

— Finding nearest neighbors

— Constructing a neigbor graph

— Varying distance

— Local connectivity

— Fuzzy area

— Merging of edges

# Step 2 - Finding a low-dimensional representation

— Minimum distance

— Minimizing the cost function

**Fig 2. UMAP workflow.**

variance ratio criterion [44].

$$VRC_k = \frac{SSB}{SSW} \times \frac{N - k}{k - 1}$$

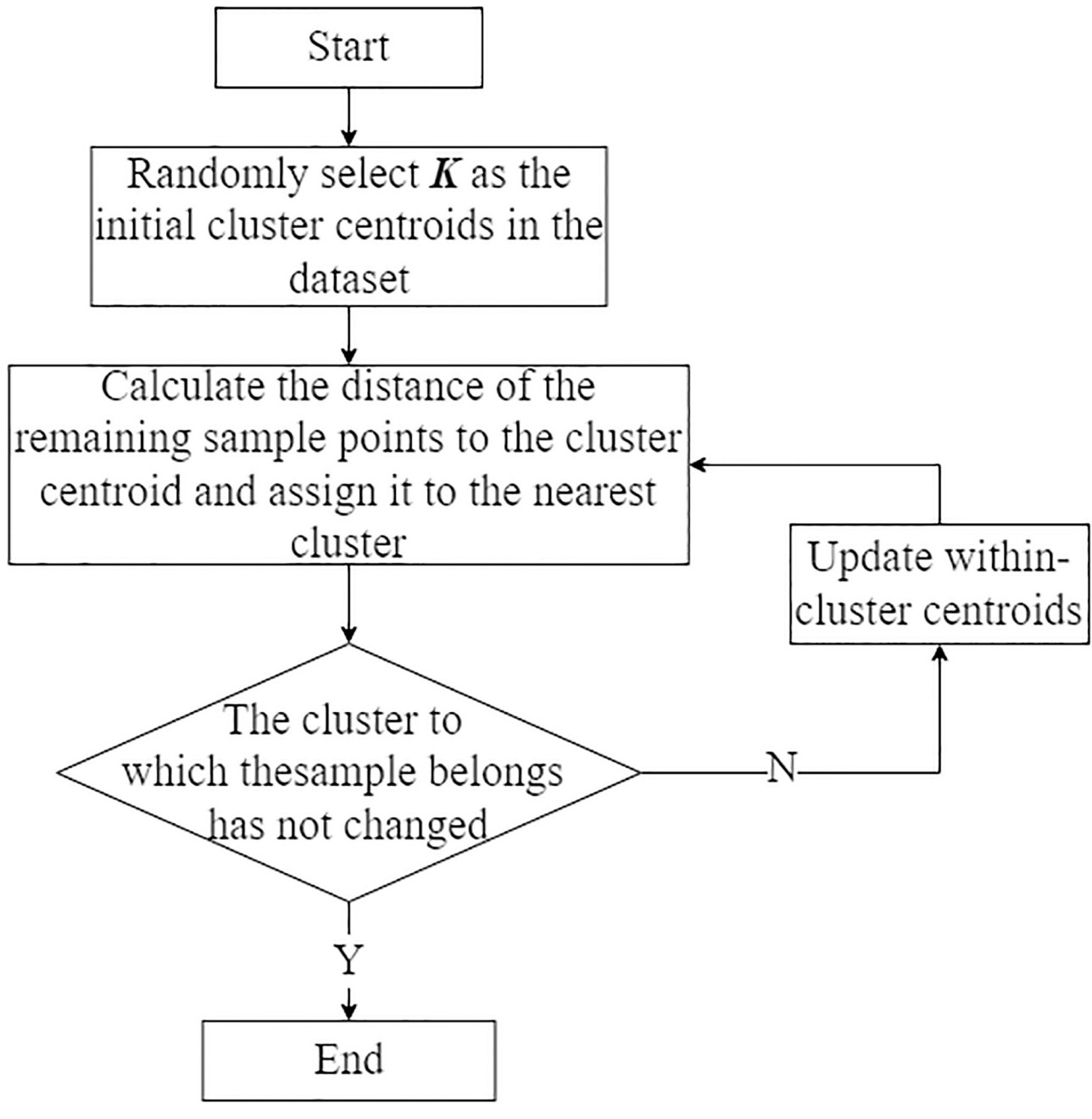

**Fig 3. K-means clustering flowchart.**

where $SSB$ is the between-class variance and defined by $SSB = \sum_{i=1}^{k} n_i \|m_i - m\|^2$, $m$ is the center point of all points, $m_i$ is the center point of a certain class; $SSW$ is the within-class variance and defined by $SSW = \sum_{i=1}^{k} \sum_{x \in C_i} \|x - m_i\|^2$; $\frac{N-k}{k-1}$ is the complexity. The larger $VRC_k$ ratio, the greater data separation.

After choosing a suitable value $K$ and completing K-means clustering, new features formed by clusters are obtained for DA. One could refer to Fig 3 for the flowchart of K-means clustering.

## Frame structure of model

We proposed a new method called TabDEG, which combines DA and TabNet and can be used for multi-classification problems such as DEGs classification and UR/DR direction prediction. PyTorch in python 3.8 is used to implement TabDEG. This model is suitable for RNA-Seq datasets with or without suitable labels and with a small number of samples, and achieves good learning results. Currently, there are no literature reports on end-to-end models based on attention mechanisms combined with DA methods for training gene expression data to predict DEGs and determine their direction of action.

TabDEG model is divided into two stages: (1)The first stage is DA and its specific structure is shown in Fig 4. The input to the first stage is two sets of vectors representing two feature sets (*N* and *T*) for each gene in cancer dataset. These two feature sets are extracted by UMAP, PCA, and K-means in the augmentation step respectively to generate new input features which are then extended to the same range or distribution. (2)The second stage is to input data into TabNet for training and prediction. The definite structure of TabNet is shown in Fig 5 and its core part consists of encoder and decoder. As shown in Fig 5, the encoder consists of Feature transformer, Attention transformer and Feature selection mask, while the decoder is relatively simple and is designed to retain only the Feature transformer part. In the stage of entire encode-decode, a special masker is used, with each dimension of the input features corresponding to a Bernoulli distribution. The parameters of Bernoulli distributions are controlled by a manually set pre-training ratio.

**DA model construction.**   In the introduction, it is pointed out that RNA-Seq datasets usually lack suitable labels, have a small number of samples, and are disproportionate to the number of genes, leading to poor performance of general DL models. To solve the problem of imbalance between the number of samples and the number of genes in RNA-Seq datasets, we used DA methods in this study.

In TabDEG, the DA approach is used to train models with aim to solve the gene classification task for cancer datasets, i.e. RNA-Seq datasets, and construct models to solve multiple classification problems such as classifying DEGs and predicting UR/DR directions. As shown in Fig 4, the features of TCGA datasets are divided into two different subsets which are labeled as *T* and *N*. For these two subsets, three DA methods (PCA, UMAP, and K-means) are respectively used to generate new valid input features (labeled as $T_1$ and $N_1$) in order to achieve effect enhancement and improve the predictive power of gene classification for model involved. Taking PCA as an example, an appropriate hyperparameter *p* for these two data subsets shoud be chosen, i.e. the remaining size after dimension reduction. Intuitively, large differences within *T* would correspond to more clusters *p* being generated while small differences within *N* would correspond to fewer clusters *p* being generated. Along this line of thought, a initial value *p* is specified and then confirm the approximate range by iterative experiments. Similarly, for UMAP and K-means, a similar approach is used to confirm the approximate range of their hyperparameters. After confirming hyperparameters of these algorithms built into these methods, effective input features could be obtained and they could typically boost effective feature information by 15% to 20%.

In the process of augmenting datasets, Fig 4 shows that all scales of raw and augmented datasets are not normalized. This management could avoid and control effect of scale variations as small as possible and each raw feature is scaled to the same range or distribution before starting training. A rank transformation could be used to smooth out the data distribution with less being affected by outliers. However, this transformation may distort the correlations and distances among and within features. In order to neutralize these two aspects, a transformation function QuantileTransformer(), which is parameter-free and based on quantile

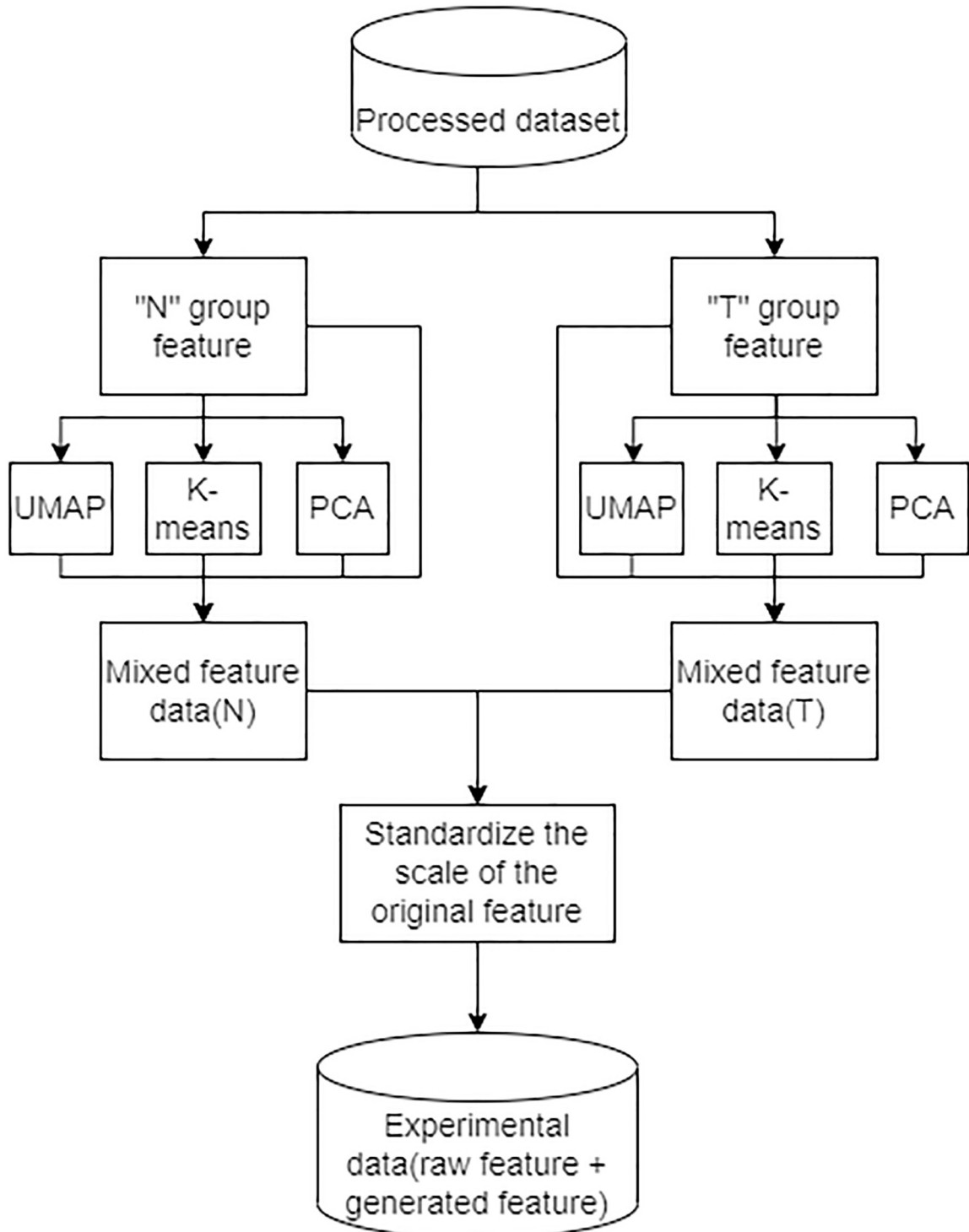

**Fig 4. DA workflow diagram.** The data is divided into two parts: disease group (*T*) and normal group (*N*). UMAP, KMEANS and PCA methods is used with aim to get mixed feature data in the process of DA. And then the data is standardized to ensure data scale consistency.

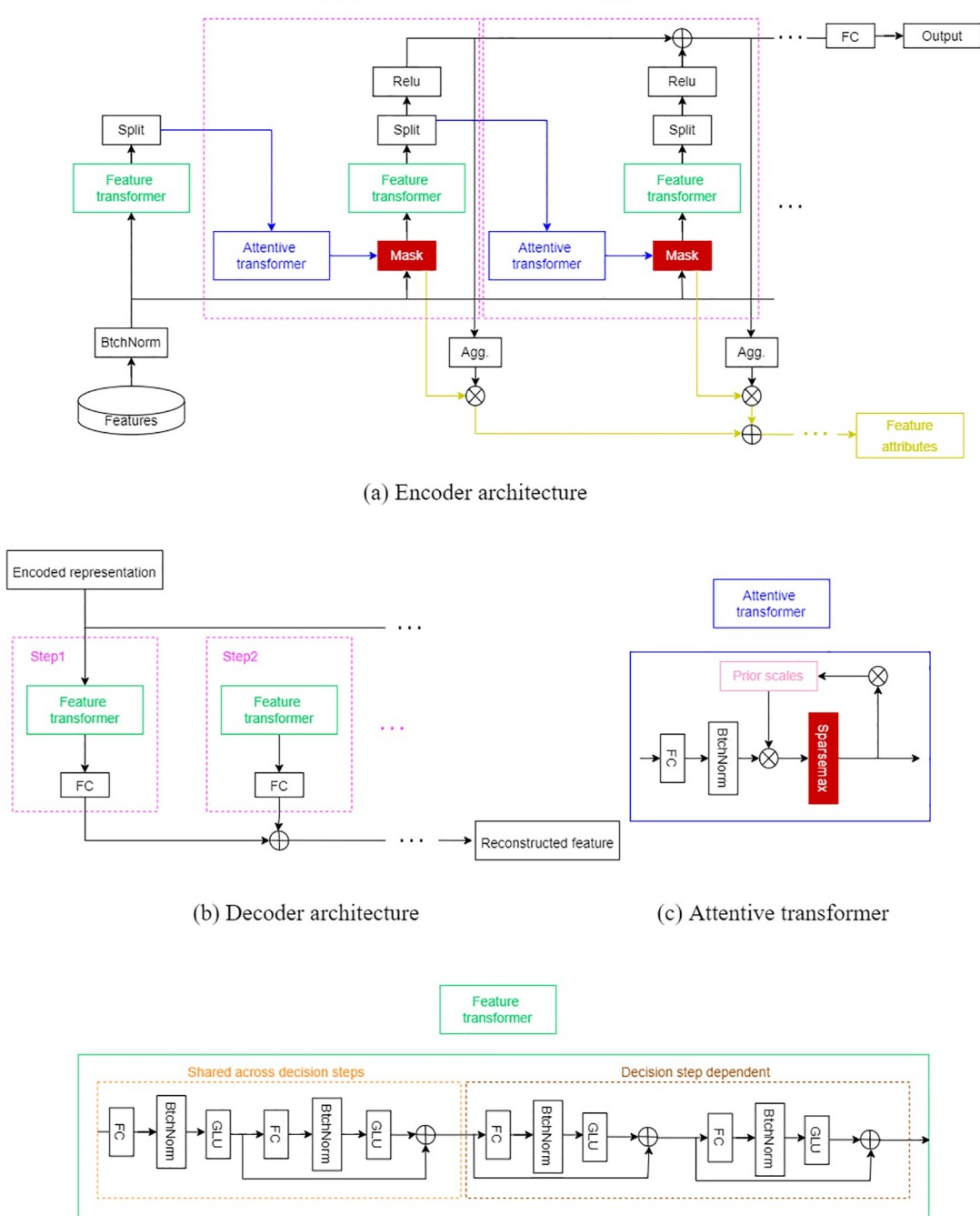

**Fig 5. TabNet structure chart.** (a) TabNet encoder, composed of a feature transformer, an attentive transformer and feature masking. A split block divides the processed representation and decomposition products obtained are used by the attentive transformer of the subsequent step as well as for the overall output. For each step, the feature selection mask provides interpretable information about the model's functionality and the masks could be aggregated to obtain global feature important attribution. (b) TabNet decoder, composed of a feature transformer block at each step. (c) Attention transformer, the features of the n_a part enter the FC layer and expand to the same dimension as the input features to facilitate subsequent attention calculations, then enter the BN layer, and then multiply with "priors" before entering the sparsemax layer. The sparsemax layer is used for feature selection and "priors" is a constant vector with all entries being 1 in the initial step, which would change in

each step. (d) Feature transformer, the output of the feature transformer is divided into two parts: n_d(n features for decision) and n_a(n features for attetion), where n_a is used for further calculation in subsequent steps and n_d is used for the final decision. (the output of the initial step has only n_a-dimensional features, rather than n_d-dimensional features to participate in the final decision).

function theory, is used to map raw feature data onto a normal distribution field. Mix transformed raw feature data with augmented new feature data together to obtain the complete experimental data ($E$). As shown in Fig 1, $E$ is divided into train data and test set in the ratio of 4 to 1 and then the train data is divided into train set and verification set in the ratio of 4 to 1. During the experiments, ten large datasets are trained and 5-fold cross-validation with 10 repetitions is executed in order to avoid bias.

**TabNet model construction.** Google Research has proposed a new high-performance, interpretable learning architecture called TabNet, which can directly train on raw tabular data without any feature engineering. In TabNet, each decision step uses sequential attention for feature selection inference and unsupervised pre-training to predict masked features, thereby improving model accuracy and interpretability. TabNet uses a single DL framework for end-to-end learning and corresponds to two interpretable points: a local interpretable point that shows the importance and combination of each input feature, and a global interpretable point that quantifies the contribution of each input feature to the output.

Feature selection mask in the stage provides interpretable information about model capabilities and this mask could be aggregated to obtain global feature important properties. As could be seen in Fig 5, the final n_d (the part that passes through the relu output) of each step is multiplied with the sparse vector output by the sparsemax layer of the Attention transformer of this round of steps and the results of the multiplication at each step are accumulated. In fact, it combines the characteristic importance of two parts which is a small integration. The output part will be filtered by the ReLU with negative number of output being directly set to 0 and then the Attention transformer would also output a mask importance. These two importances are integrated by multiplying themselves as the result of determining the feature importance of the current step to the input features. After the entire encode-decode stage, the reshaped features are obtained, the activation output of a linear layer is taken as the feature representation and Softmax() is applied to find the probability of each class in the range [0, 1].

The experiment is conducted to test the effectiveness of TabDEG model. All 10 cancer datasets are trained and used to test each dataset. Since the train data has three labels ("0", "1" and "2"), this train proess is a three-class classification problem. In the first phase of experiment, two feature sets ($N$ and $T$ features) for each gene are inputted into PCA(), KMeans(), and UMAP() respectively with aim to extract new valid features and then these two raw sets of features are inputted into transformation QuantileTransformer() for normalization. Both the normalized features and the new valid features form final experimental dataset.

In the stage of experiment, the LabelSmoothing module is built in order to reduce the weight of the real sample label category when calculating the loss function, which has the effect of suppressing overfitting and increasing the generalization ability of TabDEG model. Then TabNet runs for 100 epochs with a batch size of 512, after the outcome within LabelSmoothing module being inputted into. It uses Adam() as the optimizer to calculate the loss between a given input X and the output (y_pred) and updates the parameters according to the gradient. The pytorch-tabnet packaged in python3.8 is used to execute the experiment. The selection of hyperparameters is shown in Table 1 and the train and prediction of RNA-Seq data can be officially started after these hyperparameters are specified. For predicted class values 0, 1, 2, the input gene is accordingly classifed as DR, UR or non-DEGs. To estimate the predictive

performance of TabDEG model, the average value of four criteria (accuracy, recall, precision and F1-measure) are calculated.

## Results

In this section, the new method TabDEG proposed in this paper, data experiment details and results will be listed. The workflow of TabDEG is presented in Fig 1. It falls into two stages: the first stage includes data collection, pre-processing and labeling; the second stage consists of data augmentation, training and testing the model involved to predict DEGs and UR/DR directions. The specific architecture of neural networks involved in TabDEG is given in Fig 5 and these networks are used to train and classify empirical data studied in this paper.

Ten cancer datasets (referring to Table 2) from the TCGA project is used here to do data analysis experiment which consists of three stages. The first stage is to download these ten datasets via UCSX Xena Data Explorer and pre-process them respectively. And then the DA process is applied to these pre-processed datasets in order to encapsulate them as input data in the next stage. In second stage, the input data were fed into nueral networks listed in Fig 5 for training in order to classify DEGs and predict UR/DR directions in all datasets. In the last stage, the performance of TabDEG model is compared with other ML methods (DTC [45], CNNs [35], LSTM [36], RFC [34] and XGBoost [46]) in terms of criteria such as accuracy, recall, precision, F1-measure and ROC curve.

All through this paper, the following abbreviations are used to denote relevant datasets. The datasets after pre-processing and labeling are recorded as $R$ ("raw data") and the feature column in raw data contains normal group and tumor group. The feature column of normal group is denoted as $N$ ("normal data") while it is marked as $N_1$ after adding the feature column generated by DA process. The feature column of tumor group is denoted as $T$ ("tumor data") while it is marked as $T_1$ after adding the feature column generated by DA process. $N_1$ and $T_1$ are recombined into a new data set which is denoted as $E$ ("Experimental data"). Then experimental data $E$ is divided into train data and test set in the ratio of 4 to 1 and the train data is also divided into a train set and a validation set in the ratio of 4 to 1.

### Comparison of TabDEG performance with other ML methods

This section evaluated the performance of TabDEG and 5 other ML methods (DTC, CNNs, RFC, LSTM, and XGBoost) in predicting DEGs and UR/DR directions. Comparing using metrics such as accuracy, recall, precision, F1-measure, and ROC score, the TabDEG model and its corresponding models were trained on the training sets of all 10 cancer datasets and tested

**Table 2. Dataset abbreviations for cancer datasets.**

| Dataset abbreviation | type |
|---|---|
| BRCA | Breast invasive carcinoma |
| COAD | Colon adenocarcinoma |
| HNSC | Head and neck squamous cell carcinoma |
| KIRC | Kidney renal clear cell carcinoma |
| LUAD | Lung adenocarcinoma |
| LUSC | Lung squamous cell carcinoma |
| PRAD | Prostate adenocarcinoma |
| STAD | Stomach adenocarcinoma |
| THCA | Thyroid carcinoma |
| UCEC | Uterine Corpus endometrial carcinoma |

**Table 3. Performance of TabDEG against other five ML models on test data of all ten datasets with five-fold cross-validation being used.**

| PRECISION | TabDEG | LSTM | CNN | DTC | RCF | XGBoost | RECALL | TabDEG | LSTM | CNN | DTC | RCF | XGBoost |
|---|---|---|---|---|---|---|---|---|---|---|---|---|---|
| BRCA | 0.93 | 0.9 | 0.79 | 0.74 | 0.75 | 0.91 | BRCA | 0.93 | 0.82 | 0.74 | 0.72 | 0.7 | 0.88 |
| COAD | 0.94 | 0.92 | 0.92 | 0.79 | 0.76 | 0.92 | COAD | 0.94 | 0.87 | 0.86 | 0.78 | 0.67 | 0.89 |
| HNSC | 0.93 | 0.91 | 0.78 | 0.75 | 0.74 | 0.9 | HNSC | 0.93 | 0.89 | 0.7 | 0.74 | 0.66 | 0.88 |
| KIRC | 0.94 | 0.94 | 0.85 | 0.77 | 0.76 | 0.92 | KIRC | 0.94 | 0.88 | 0.73 | 0.77 | 0.78 | 0.91 |
| LUAD | 0.94 | 0.91 | 0.71 | 0.78 | 0.75 | 0.92 | LUAD | 0.94 | 0.84 | 0.74 | 0.77 | 0.75 | 0.89 |
| LUSC | 0.96 | 0.88 | 0.92 | 0.81 | 0.79 | 0.93 | LUSC | 0.96 | 0.9 | 0.87 | 0.81 | 0.8 | 0.93 |
| PRAD | 0.93 | 0.9 | 0.86 | 0.72 | 0.78 | 0.91 | PRAD | 0.93 | 0.84 | 0.78 | 0.72 | 0.54 | 0.84 |
| STAD | 0.93 | 0.9 | 0.88 | 0.74 | 0.72 | 0.9 | STAD | 0.93 | 0.89 | 0.8 | 0.73 | 0.65 | 0.87 |
| THCA | 0.93 | 0.91 | 0.76 | 0.71 | 0.77 | 0.9 | THCA | 0.93 | 0.87 | 0.83 | 0.71 | 0.57 | 0.82 |
| UCEC | 0.94 | 0.89 | 0.9 | 0.78 | 0.74 | 0.91 | UCEC | 0.94 | 0.83 | 0.84 | 0.78 | 0.73 | 0.91 |
| **F1-SCORE** | **TabDEG** | **LSTM** | **CNN** | **DTC** | **RCF** | **XGBoost** | **ACCURACY** | **TabDEG** | **LSTM** | **CNN** | **DTC** | **RCF** | **XGBoost** |
| BRCA | 0.93 | 0.85 | 0.72 | 0.73 | 0.69 | 0.89 | BRCA | 0.93 | 0.89 | 0.74 | 0.79 | 0.79 | 0.92 |
| COAD | 0.94 | 0.89 | 0.88 | 0.78 | 0.67 | 0.9 | COAD | 0.94 | 0.91 | 0.86 | 0.81 | 0.75 | 0.92 |
| HNSC | 0.93 | 0.9 | 0.7 | 0.74 | 0.65 | 0.89 | HNSC | 0.93 | 0.91 | 0.7 | 0.78 | 0.74 | 0.9 |
| KIRC | 0.94 | 0.9 | 0.7 | 0.77 | 0.76 | 0.91 | KIRC | 0.94 | 0.91 | 0.73 | 0.8 | 0.81 | 0.92 |
| LUAD | 0.94 | 0.86 | 0.71 | 0.77 | 0.74 | 0.9 | LUAD | 0.94 | 0.88 | 0.74 | 0.81 | 0.8 | 0.92 |
| LUSC | 0.96 | 0.89 | 0.89 | 0.81 | 0.78 | 0.93 | LUSC | 0.96 | 0.88 | 0.87 | 0.81 | 0.8 | 0.93 |
| PRAD | 0.93 | 0.86 | 0.79 | 0.72 | 0.51 | 0.87 | PRAD | 0.93 | 0.92 | 0.78 | 0.83 | 0.79 | 0.92 |
| STAD | 0.93 | 0.89 | 0.81 | 0.74 | 0.66 | 0.89 | STAD | 0.93 | 0.91 | 0.8 | 0.77 | 0.74 | 0.9 |
| THCA | 0.93 | 0.89 | 0.78 | 0.71 | 0.6 | 0.86 | THCA | 0.93 | 0.94 | 0.83 | 0.85 | 0.84 | 0.93 |
| UCEC | 0.94 | 0.85 | 0.84 | 0.78 | 0.72 | 0.91 | UCEC | 0.94 | 0.85 | 0.84 | 0.8 | 0.75 | 0.91 |

using 5-fold cross-validation. The experimental results showed that, as shown in Table 3, the scores of TabDEG exceeded 93% for each dataset, with a score of 96% for the LUSC dataset. Compared with the 5 control ML models, TabDEG scores at least 2% higher. In terms of ROC scores, as shown in Fig 6 (only two datasets are presented in the text, while the rest are presented in the S1 File), the scores for all 10 cancer datasets were greater than 0.9. Compared with other ML methods, TabDEG performed better on most datasets, while CNNs and LSTM showed more fluctuation in their results. The experimental results show that The TabDEG model is better than the control five ML models and can effectively classify DEGs and predict UR/DR genes.

## GO enrichment analysis of predicted UR and DR genes

After classifying genes into DEGs and their UR/DR directions using our TabDEG model, we evaluated the GO enrichment of the predicted UR and DR genes using ToppGene Suite. As shown in the Table 4, the predicted UR and DR genes were enriched with some common GO terms associated with carcinogenesis.

According to the data provided and GO ID/attribute, we can see that the predicted UR and DR gene mappings in the BRCA and UCEC datasets are mainly related to processes such as tumor cell proliferation, survival, invasion, and metastasis [47]. In the BRCA dataset, common GO terms related to cancer include tumor cell proliferation, survival, invasion, and metastasis, such as the G protein-coupled receptor signaling pathway (such as GO:0007186) which usually plays an important role in tumor cell proliferation and infiltration; pathways related to cell death and immune response (such as GO:0006959 and GO:0031640) are related to the anti-tumor effect and immune escape of tumor cells; pathways related to extracellular matrix

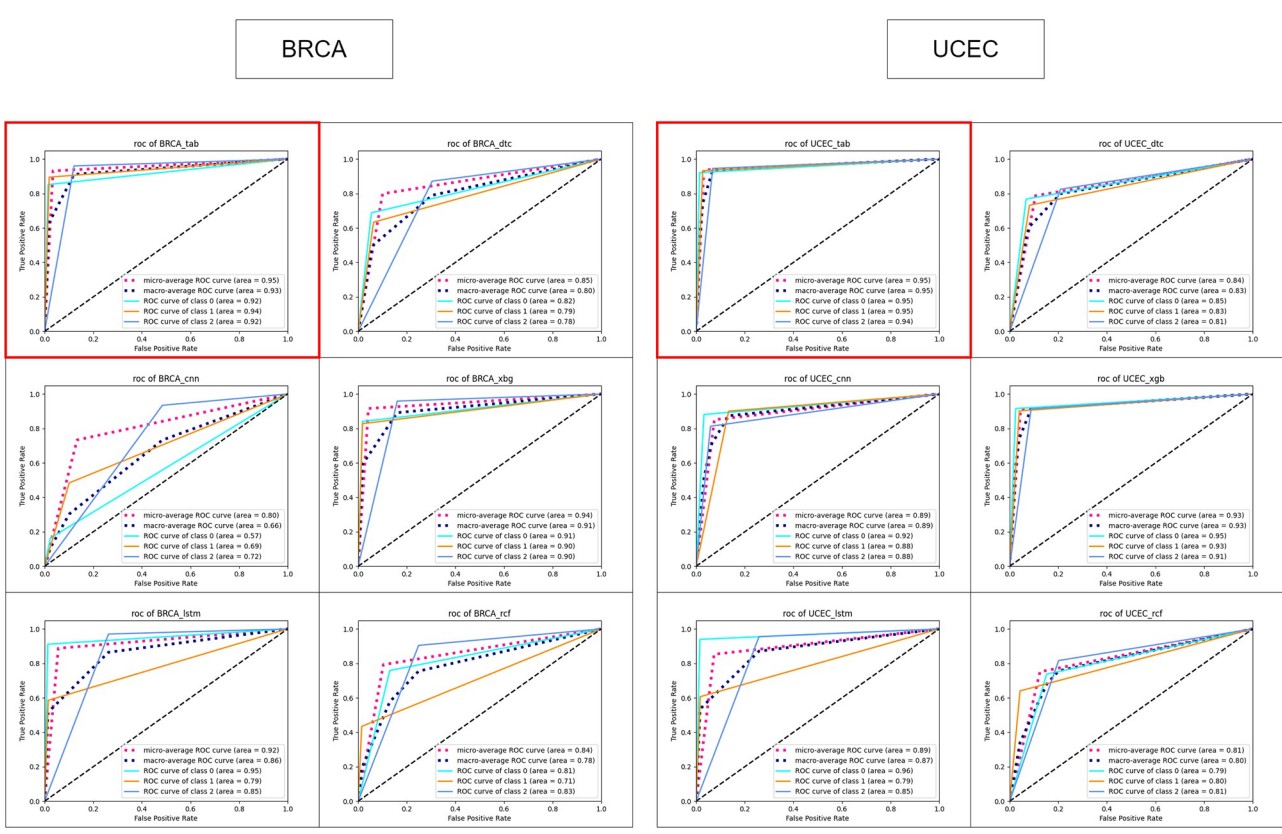

**Fig 6. ROC curves with scores for different methods across ten datasets.**

degradation and cell movement (such as GO:0099537 and GO:0045165) are related to the invasion and metastasis of tumor cells [48].

For the UCEC dataset, the GO terms mapped by the predicted UR and DR genes mainly involve aspects such as cell adhesion and connection, cell signal transduction, and enzyme-linked receptor protein signaling, and the abnormal activation or inhibition of these pathways is closely related to the occurrence and development of various cancers [49]. For example, the cell-cell adhesion protein E-cadherin plays an important role in the adhesion and metastasis of cancer cells, while pathways such as the RAS-MAPK pathway and the PI3K-AKT-mTOR pathway are abnormally activated in many tumors, directly participating in the growth and invasion of tumor cells [50]. The significant enrichment of these GO terms suggests that these biological processes may play a key role in tumors and may become targets for drug development.

### Pathway enrichment analysis of predicted UR and DR genes

We performed pathway enrichment analysis on the predicted UR and DR genes in the biological test data of the BRCA and UCEC datasets. We reported 13 important pathways related to the progress of various cancer datasets, all of which are sourced from the predicted UR and DR genes of BRCA and UCEC in S2 File. We will discuss the DEGs that are repeatedly mapped to multiple pathways (using the BRCA data set as an example) in the following paragraph. In Fig 7, we show the UR/DR genes in the BRCA dataset that are repeatedly mapped to multiple pathways [51, 52].

**Table 4. Datasets GO ID/attribute p-value q-value.**

| Datasets | GO ID/attribute | p-value | q-value |
|---|---|---|---|
| BRCA | G protein-coupled receptor signaling pathway | 1.74E-08 | 1.02E-04 |
| | cell fate commitment | 2.32E-04 | 1.00E+00 |
| | monocarboxylic acid metabolic process | 2.43E-04 | 1.00E+00 |
| | negative regulation of cell development | 3.80E-04 | 1.00E+00 |
| | modulation of chemical synaptic transmission | 1.53E-04 | 8.98E-01 |
| | humoral immune response | 1.74E-07 | 1.02E-03 |
| | killing of cells of another organism | 6.44E-07 | 3.78E-03 |
| | trans-synaptic signaling | 1.80E-06 | 1.06E-02 |
| | positive regulation of protein kinase A signaling | 9.52E-05 | 2.97E-01 |
| UCEC | antimicrobial humoral response | 1.07E-08 | 7.09E-05 |
| | humoral immune response | 4.61E-07 | 3.06E-03 |
| | keratinocyte differentiation | 4.68E-06 | 3.10E-02 |
| | meiotic cell cycle | 4.50E-05 | 2.98E-01 |
| | antibacterial humoral response | 6.25E-05 | 4.14E-01 |
| | skin development | 6.64E-05 | 4.40E-01 |
| | nuclear division | 1.17E-04 | 7.78E-01 |
| | angiogenesis | 1.05E-08 | 5.92E-05 |
| | regulation of angiogenesis | 7.10E-05 | 4.00E-01 |
| | positive regulation of cell differentiation | 1.67E-04 | 9.44E-01 |
| | ameboidal-type cell migration | 1.10E-03 | 1.00E+00 |
| | regulation of cell migration | 4.31E-05 | 2.43E-01 |
| | epithelial cell proliferation | 5.40E-04 | 1.00E+00 |
| | regulation of transporter activity | 7.10E-05 | 4.00E-01 |
| | G protein-coupled receptor signaling pathway | 1.38E-04 | 7.81E-01 |

UR genes: HTR2A is involved in tumor cell proliferation, invasion, and metastasis and is associated with the occurrence and progression of breast cancer [53]; CCL11 is related to cancer-related inflammation [54]; PTHLH is also considered a breast cancer growth factor [55]; PROX1 participates in breast cancer formation and development by inhibiting angiogenesis and regulating epithelial cell proliferation [56]; NKX6–1 can inhibit the proliferation and invasion of breast cancer cells [57]; IFNG is an important factor in anti-tumor immune response [58]; MMP9 is a protease that promotes tumor invasion and metastasis [59]; DRD2 may affect tumor growth and metastasis by regulating the proliferation and migration of tumor cells in neurons [60]; KIT is involved in the proliferation and invasion of tumor cells [61].

DR genes: COMP can promote the proliferation, invasion, and metastasis of breast cancer cells and has a certain impact on the occurrence and progression of breast cancer [62]; ADIPOQ, as an anti-tumor protein, plays an important role in regulating the apoptosis, metabolism, and immune response of tumor cells, and its level is closely related to the prognosis of breast cancer patients [63].

These UR/DR genes identified by our model play important roles in complex signal transduction networks, participating in the regulation and cross-reaction of multiple signaling pathways. Further research on these genes can reveal their mechanisms and key nodes in the entire signal transduction network, providing a more comprehensive and detailed understanding for the development of new therapeutic strategies.

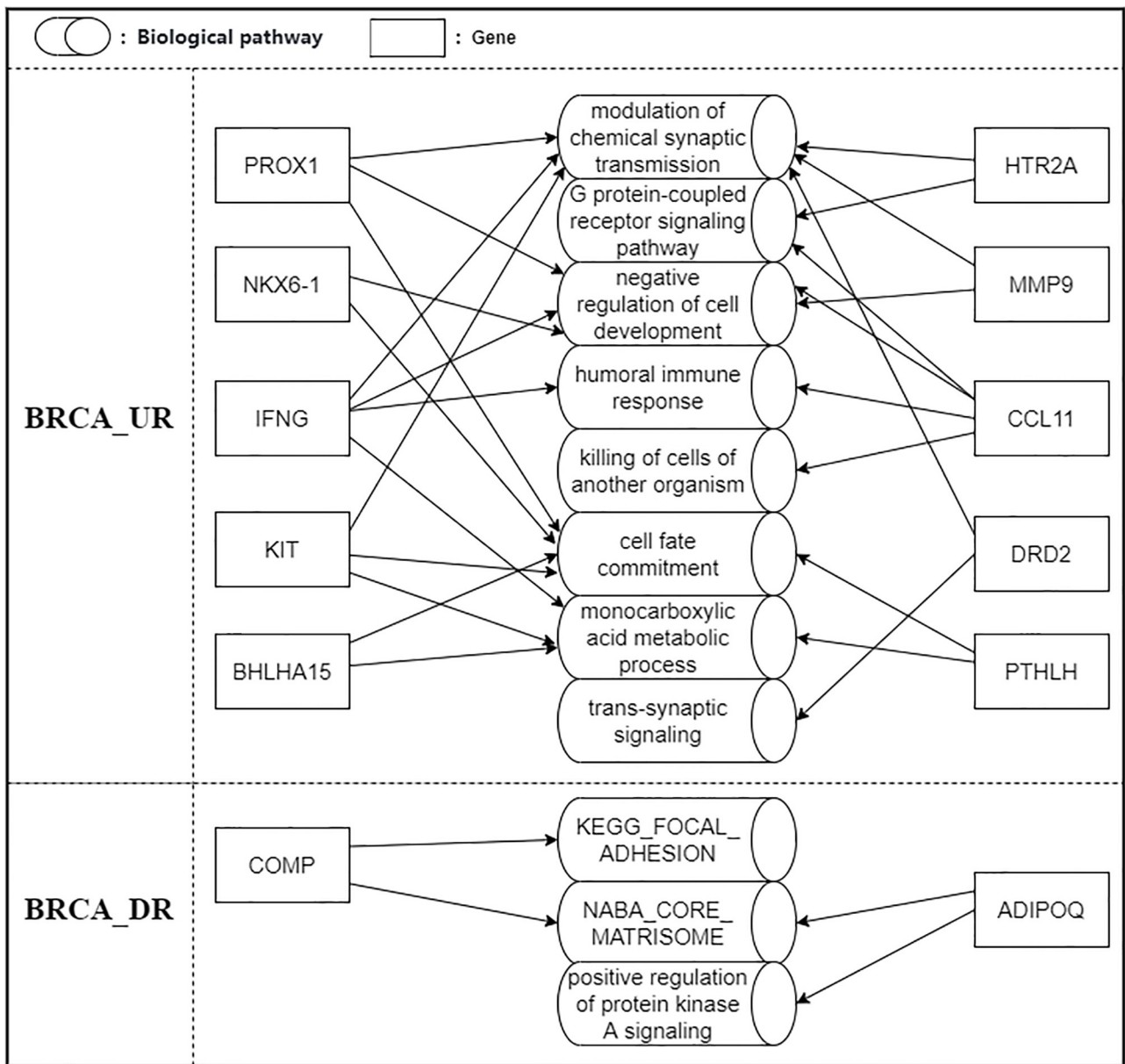

**Fig 7. Pathways mapped from predicted UR and DR genes of BRCA.**

## Discussion & conclusion

With the continuous development of NGS technology, RNA-seq has become an important tool for exploring cellular heterogeneity and disease states. However, there is currently a lack of classification methods suitable for all large-scale datasets. Neural network models are not limited by data distribution and are therefore suitable for classifying all RNA-Seq data. DL has demonstrated strong performance advantages in fields such as bioinformatics and will further explore biological problems such as gene screening in precision medicine in the future. However, applying DL to classify RNA-Seq data remains highly challenging, with one reason being the significant imbalance between sample size and gene features. In addition, these data are

typically unstructured, so commonly used neural network techniques still have limitations in gene expression data.

This paper proposes a model called TabDEG, which combines DA and TabNet, aiming to solve multiple classification problems, such as classifying DEGs and predicting UR/DR direction. The model can be generalized to RNA-Seq datasets without restrictions on sample size and appropriate labels, and achieve good learning results. Experimental results show that the TabDEG model performs better than (at least comparable to) corresponding machine learning models, effectively classifying DEGs and predicting UR/DR direction from the test sets of these TCGA cancer datasets. We validated the biological enrichment of predicted UR and DR genes in BRCA and UCEC datasets, including GO and pathway enrichment. We found that predicted UR and DR genes were significantly enriched in cancer-related GO terms with significant p-values and q-values. Similarly, in terms of biological pathways, we found that predicted UR and DR genes were enriched in pathways related to breast cancer, such as the NABA MATRISOME and NABA CORE MATRISOME pathways. The pathways mapped by predicted UR and DR genes from UCEC datasets also played an important role in carcinogenesis, such as KEGG metabolic disease-related pathways.

The proposed TabDEG model provides a new method for predicting DEGs and their UR/DR direction from both trained and untrained datasets using logFC values and statistical knowledge. Through downstream analysis of predicted UR and DR genes, we gained insight into potential mechanisms and helped identify regulatory factors for breast cancer and endometrial cancer. Therefore, by predicting and classifying DEGs and their UR/DR direction, TabDEG can help explore potential biomarkers from other RNA-seq datasets.

## Supporting information

**S1 File. ROC curves with scores for different methods across ten datasets.**
(PDF)

**S2 File. Mapped predicted genes in cancer pathways.**
(PDF)

## Author Contributions

**Methodology:** Yinghua Jin.

**Writing – original draft:** Sifan Feng.

**Writing – review & editing:** Zhenyou Wang, Yinghua Jin, Shengbin Xu.

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
