## [Decision Letter · Decision Letter 0]

2 Jan 2024

PONE-D-23-30756TabDEG: Classifying differentially expressed genes from RNA-seq data based on feature extraction and deep learning frameworkPLOS ONE

Dear Dr. JIN,

Thank you for submitting your manuscript to PLOS ONE. After careful consideration, we feel that it has merit but does not fully meet PLOS ONE’s publication criteria as it currently stands. Therefore, we invite you to submit a revised version of the manuscript that addresses the points raised during the review process.

We look forward to receiving your revised manuscript.

Kind regards,

Divijendra Natha Reddy Sirigiri

Academic Editor

PLOS ONE

Journal Requirements:

Did you know that depositing data in a repository is associated with up to a 25% citation advantage (https://doi.org/10.1371/journal.pone.0230416)? If you’ve not already done so, consider depositing your raw data in a repository to ensure your work is read, appreciated and cited by the largest possible audience. You’ll also earn an Accessible Data icon on your published paper if you deposit your data in any participating repository (https://plos.org/open-science/open-data/#accessible-data).

3. Please note that PLOS ONE has specific guidelines on code sharing for submissions in which author-generated code underpins the findings in the manuscript. In these cases, all author-generated code must be made available without restrictions upon publication of the work. 

Please review our guidelines at https://journals.plos.org/plosone/s/materials-and-software-sharing#loc-sharing-code and ensure that your code is shared in a way that follows best practice and facilitates reproducibility and reuse.

4. Please update your submission to use the PLOS LaTeX template. The template and more information on our requirements for LaTeX submissions can be found at http://journals.plos.org/plosone/s/latex. 

6. We are unable to open your Supporting Information file [figures.rar]. Please kindly revise as necessary and re-upload.

Reviewers' comments:

Reviewer's Responses to Questions

**Comments to the Author**

1. Is the manuscript technically sound, and do the data support the conclusions?

Reviewer #1: Yes

Reviewer #2: Yes

2. Has the statistical analysis been performed appropriately and rigorously? 

Reviewer #1: Yes

Reviewer #2: Yes

3. Have the authors made all data underlying the findings in their manuscript fully available?

Reviewer #1: Yes

Reviewer #2: Yes

4. Is the manuscript presented in an intelligible fashion and written in standard English?

Reviewer #1: No

Reviewer #2: No

5. Review Comments to the Author

Reviewer #1: It's my honor to receive the manuscript entitled 'TabDEG: Classifying differentially expressed genes from RNA-seq data based on feature extraction and deep learning framework'. Feng et al offered a TabDEG model which combines DA and TabNet, aiming to solve multiple classification problems, such as classifying DEGs and predicting UR/DR direction.However, it is highly recommended that you address the issues listed before submitting. Please see comments below:

1. The authors should mention in detail how they selected the 10 different tumor types characteristic (Age, gender, stage, TNM, et al) for the experiment and discuss the potential bias of the results.

2. The author should justify using logfc > 1 as a threshold for DEGs identification.

3.Necessary references should be added, such as the section on "Path enrich analysis of predicted UR and DR genes".

4.English needs to be improved by English editing service or a native English speaker.

Reviewer #2: The work is interesting. This paper designs framework for predicting DEGs. Although the promising results have been achieved, several major concerns in the current manuscript show that it appears that publication in any form would be premature at this time.

1. The relationship between the data downloaded from UCSX Xena Data Browser (https : //xenabrowser.net/datapages/) and the data of TCGA should be listed.

2.” Import the ”tidyverse” package and then start pre-processing the gene expression

data. RNA transcripts with an expression level greater than 1 are screened out and a

total of 17983 RNA transcripts are screened. The screen-obtained datasets are divided into T and N, which are used as control conditions in next step.” Here, how to divide the two datasets ”T” and “N”?

3. The readability of writing is poor, which makes the manuscript is difficult to be understood.

4. Tables 3 and 4 can be combined and shown graphically.

5. Including the benchmarking algorithms, what is the input of the algorithms? What kind of feature extraction method is used?

6. It was suggested that authors should share the code and data in this work.

6. PLOS authors have the option to publish the peer review history of their article (what does this mean?). If published, this will include your full peer review and any attached files.

Reviewer #1: **Yes: **Qinglu Wang

Reviewer #2: No

---

## [Author Response · Author response to Decision Letter 0]

20 Feb 2024

Dear Editors and Reviewers:

Thank you for your letter and for the reviewers’ comments concerning our manuscript entitled “TabDEG: Classifying differentially expressed genes from RNA-seq data based on feature extraction and deep learning framework” (PONE-D-23-30756). Those comments are all valuable and very helpful for revising and improving our paper, as well as the important guiding significance to our researches. We have studied comments carefully and have made correction which we hope meet with approval. The maincorrections in the paper and the responds to the reviewer's comments are as flowing:

Responds to the reviewer's comments:

Reviewer#1:

1.Response to comment: The authors should mention in detail how they selected the 10 different tumor types characteristic (Age, gender, stage, TNM, et al) for the experiment and discuss the potential bias of the results.

Response: 

The TCGA dataset selected in this paper comes from a variety of samples from patients with different regions, races and clinical characteristics, and thus contains different genomic information, which helps to reduce group bias and improve the representativeness of the results. Meanwhile, different tumor types may have different characteristics, development patterns, and treatments, so by using datasets of multiple tumor types, the over-reliance on specific tumor types can be reduced, and the generalization ability of the algorithm can be improved, making it more robust in dealing with unknown or novel tumor types. Relevant revisions we have marked in yellow color in the "Data Collection and Preprocessing" section of the revised manuscript.

2.Response to comment:The author should justify using logfc > 1 as a threshold for DEGs identification.

Responsc:

LogFC > 1 indicates at least a 2-fold change in the expression level of a gene between two conditions, and such a change is considered to be relatively large to provide a higher significance difference and thus more likely to reflect biological importance. In many related studies, logFC > 1 is considered a reasonable threshold for screening out DEGs of functional and biological importance. Relevant revisions we have marked in yellow color in the "Data Collection and Preprocessing" section of the revised manuscript.

3.Response to comment:Necessary references should be added, such as the section on "Path enrich analysis of predicted UR and DR genes".

Responsc:

We sincerely appreciate the valuable comments. We have checked the literature carefully and added more references into the section on "Path enrich analysis of predicted UR and DR genes" part in the revised manuscript.

4.Response to comment:English needs to be improved by English editing service or a native English speaker.

Responsc: 

Thanks for your suggestion. We have tried our best to polish the language in the revised manuscript.

Reviewer#2:

1.Response to comment: The relationship between the data downloaded from UCSX Xena Data Browser (https : //xenabrowser.net/datapages/) and the data of TCGA should be listed.

Response:

The UCSX Xena Data Browser serves as a data browser that provides access to and analyzes TCGA data. Relevant revisions we have marked in red color in the "Data Collection and Preprocessing" section of the revised manuscript.

2.Response to comment: ” Import the ”tidyverse” package and then start pre-processing the gene expression data. RNA transcripts with an expression level greater than 1 are screened out and a

total of 17983 RNA transcripts are screened. The screen-obtained datasets are divided into T and N, which are used as control conditions in next step.” Here, how to divide the two datasets ”T” and “N”?

Response: 

The tumor and normal groups will be labeled "01A" and "11A" respectively in the column names of the TCGA dataset, and based on this the dataset will be categorized into "T" and "N" groups. Relevant revisions we have marked in red color in the "Data Collection and Preprocessing" section of the revised manuscript.

3.Response to comment: The readability of writing is poor, which makes the manuscript is difficult to be understood.

Response: 

Thanks for your suggestion. We have tried our best to polish the language in the revised manuscript.

4.Response to comment: Tables 3 and 4 can be combined and shown graphically.

Response: 

We have completed the relevant revisions in the Table 3 of the revised manuscript.

5.Response to comment: Including the benchmarking algorithms, what is the input of the algorithms? What kind of feature extraction method is used?

Response: 

The input to the algorithm is RNA-seq counts data, and the feature extraction uses a combination of pca, k-means and umap for dimensionality reduction extraction.

6.Response to comment: It was suggested that authors should share the code and data in this work.

Response: 

We will share the code and data from this work in the Supporting Information.

Special thanks to you for your good comments.

---

## [Decision Letter · Decision Letter 1]

1 Apr 2024

PONE-D-23-30756R1TabDEG: Classifying differentially expressed genes from RNA-seq data based on feature extraction and deep learning frameworkPLOS ONE

Dear Dr. JIN,

Thank you for submitting your manuscript to PLOS ONE. After careful consideration, we feel that it has merit but does not fully meet PLOS ONE’s publication criteria as it currently stands. Therefore, we invite you to submit a revised version of the manuscript that addresses the points raised during the review process.

We look forward to receiving your revised manuscript.

Kind regards,

Divijendra Natha Reddy Sirigiri

Academic Editor

PLOS ONE

Journal Requirements:

Reviewers' comments:

Reviewer's Responses to Questions

**Comments to the Author**

1. If the authors have adequately addressed your comments raised in a previous round of review and you feel that this manuscript is now acceptable for publication, you may indicate that here to bypass the “Comments to the Author” section, enter your conflict of interest statement in the “Confidential to Editor” section, and submit your "Accept" recommendation.

Reviewer #2: (No Response)

2. Is the manuscript technically sound, and do the data support the conclusions?

Reviewer #2: Yes

3. Has the statistical analysis been performed appropriately and rigorously? 

Reviewer #2: Yes

4. Have the authors made all data underlying the findings in their manuscript fully available?

Reviewer #2: Yes

5. Is the manuscript presented in an intelligible fashion and written in standard English?

Reviewer #2: Yes

6. Review Comments to the Author

Reviewer #2: Authors should share the code and data in this work，it can be posted on GitHub instead of screenshots in supplementary materials

7. PLOS authors have the option to publish the peer review history of their article (what does this mean?). If published, this will include your full peer review and any attached files.

Reviewer #2: No

---

## [Author Response · Author response to Decision Letter 1]

10 May 2024

Dear Editors and Reviewers:

Thank you for your letter and for the reviewers’ comments concerning our manuscript entitled “TabDEG: Classifying differentially expressed genes from RNA-seq data based on feature extraction and deep learning framework” (PONE-D-23-30756R1). Those comments are all valuable and very helpful for revising and improving our paper, as well as the important guiding significance to our researches. We have studied comments carefully and have made correction which we hope meet with approval. The maincorrections in the paper and the responds to the reviewer's comments are as flowing:

Responds to the reviewer's comments:

Reviewer#2:

Authors should share the code and data in this work，it can be posted on GitHub instead of screenshots in supplementary materials

Response: 

Thanks for the correction, we have posted the code and data in this work on GitHub, which can be viewed at the link: https://github.com/xueyupi/my_tabdeg

Special thanks to you for your good comments.

---

## [Editor Report · Decision Letter 2]

6 Jun 2024

TabDEG: Classifying differentially expressed genes from RNA-seq data based on feature extraction and deep learning framework

PONE-D-23-30756R2

Dear Dr. JIN,

We’re pleased to inform you that your manuscript has been judged scientifically suitable for publication and will be formally accepted for publication once it meets all outstanding technical requirements.

Kind regards,

Divijendra Natha Reddy Sirigiri

Academic Editor

PLOS ONE
---

## [Editor Report · Acceptance letter]

11 Jul 2024

PONE-D-23-30756R2 

PLOS ONE

Dear Dr. Jin, 

I'm pleased to inform you that your manuscript has been deemed suitable for publication in PLOS ONE. Congratulations! Your manuscript is now being handed over to our production team.

Kind regards, 

on behalf of

Dr. Divijendra Natha Reddy Sirigiri 

Academic Editor

PLOS ONE